# Unravelling the complex causal effects of substance use behaviours on common diseases

Angli Xue [1,2,3], Zhihong Zhu [1,4], Huanwei Wang [1], Longda Jiang[1], Peter M. Visscher [1], Jian Zeng [1] & Jian Yang [1,5,6] ✉

## Abstract

**Background** Substance use behaviours (SUB) including smoking, alcohol consumption, and coffee intake are associated with many health outcomes. However, whether the health effects of SUB are causal remains controversial, especially for alcohol consumption and coffee intake.

**Methods** In this study, we assess 11 commonly used Mendelian Randomization (MR) methods by simulation and apply them to investigate the causal relationship between 7 SUB traits and health outcomes. We also combine stratified regression, genetic correlation, and MR analyses to investigate the dosage-dependent effects.

**Results** We show that smoking initiation has widespread risk effects on common diseases such as asthma, type 2 diabetes, and peripheral vascular disease. Alcohol consumption shows risk effects specifically on cardiovascular diseases, dyslipidemia, and hypertensive diseases. We find evidence of dosage-dependent effects of coffee and tea intake on common diseases (e.g., cardiovascular disease and osteoarthritis). We observe that the minor allele effect of rs4410790 (the top signal for tea intake level) is negative on heavy tea intake ($\hat{b}_{GWAS} = -0.091, s.e. = 0.007, P = 4.90 \times 10^{-35}$) but positive on moderate tea intake ($\hat{b}_{GWAS} = 0.034, s.e. = 0.006, P = 3.40 \times 10^{-8}$), compared to the non-tea-drinkers.

**Conclusion** Our study reveals the complexity of the health effects of SUB and informs design for future studies aiming to dissect the causal relationships between behavioural traits and complex diseases.

## Plain language summary

Many people smoke or consume alcohol, coffee and tea. The relationship between using these types of substance and the development of different diseases is not well understood. Previous studies have suggested that differences in genetics, i.e. inherited characteristics, could have an impact on how each substance impacts a particular person's health. We used a method called Mendelian Randomization to look at the impact of consuming tobacco, alcohol, coffee and tea on the development of various common diseases using genetic information. We found that relationships were complicated and many were dosage-dependent, but that consumption of a large amount of all substances tended to have negative health impacts regardless of lifestyle, behavioural or inherited characteristics.

The consumption of various substances, including tobacco, alcohol, and drugs, is known as substance use behaviours (SUB). These behaviours can potentially lead to dependence or disorders related to substance use, which can substantially affect human health[1–4]. Tobacco smoking is linked to ~6 million deaths globally every year[5], and is also a major contributor to chronic respiratory diseases in the UK[6]. Global alcohol consumption is associated with ~3 million deaths annually[7], and individuals who consume alcohol excessively may face a range of health complications. Meanwhile, beverages like coffee and tea, known to contain stimulants such as caffeine, are consumed widely but are subject to limited regulatory oversight. Long-term and

heavy consumption of coffee could result in caffeine dependence, and discontinuation may lead to withdrawal symptoms such as fatigue, difficulty concentrating, and muscle pain[8]. For specific substances like alcohol and coffee, their potential benefits are still controversial and under heavy debate[9–11]. Understanding the causal effects of SUB on common diseases is essential to guide disease prevention and intervention.

Observational studies have provided evidence for associations between SUB and common diseases such as the associations between smoking and lung cancer[12] and between alcohol use and breast cancer[13]. However, observational studies are vulnerable to confounding effects and reverse

[1]Institute for Molecular Bioscience, The University of Queensland, Brisbane, QLD 4072, Australia. [2]Garvan-Weizmann Centre for Cellular Genomics, Garvan Institute of Medical Research, Sydney, NSW 2010, Australia. [3]School of Biomedical Sciences, University of New South Wales, Sydney, NSW 2052, Australia. [4]National Centre for Register-Based Research, Aarhus University, Aarhus V 8210, Denmark. [5]School of Life Sciences, Westlake University, Hangzhou, Zhejiang 310024, China. [6]Westlake Laboratory of Life Sciences and Biomedicine, Hangzhou, Zhejiang 310024, China. ✉e-mail: jian.yang@westlake.edu.cn

causality, which could lead to biased effect estimates. Randomized Controlled Trial (RCT) is considered as a gold standard to test for causality, but it could be expensive and time-consuming, sometimes unethical or impractical. Mendelian Randomization (MR) is a statistical method to estimate the causal effect of a modifiable exposure on a health outcome using the exposure-associated genetic variants (e.g. SNPs) as instrumental variables (IVs)[14]. Recent MR studies have provided evidence for putative causal associations between smoking behaviour and obesity[15], between alcohol intake and cardiovascular disease[16], and between smoking initiation and schizophrenia[17]. The validity of the MR framework relies on several core assumptions[14] (e.g., valid IVs should only affect outcome via exposure), while in real data analysis, those assumptions are not always fulfilled (e.g., IVs can have direct effects on outcome, a phenomenon dubbed horizontal pleiotropy, commonly seen in genetic studies). Although many MR methods have been developed to deal with pleiotropy[18–20], the extent to which these methods are robust to horizontal pleiotropy remains elusive.

In this study, we investigate the putative causal associations between seven SUB traits, namely smoking initiation, current smoking, past smoking, smoking cessation, alcohol consumption, coffee intake and tea intake, and a range of common diseases. Summary statistics of these traits are either from published genome-wide association studies (GWAS; sample size $n = 16{,}731–547{,}261$) or in-house GWAS using the UK Biobank (UKB)[21] data ($n = 208{,}988–454{,}648$). To ensure robust and reliable estimates of the causal effects, we calibrate 11 commonly used MR methods by simulation before applying them to real data. We also investigate whether the causal effect estimates could be confounded by socioeconomic status (SES) and physical activity (PA), aiming to determine if the IVs have effects on the outcome via pathways other than SUB traits. Our study identifies putative causal links between SUB and common diseases, highlights the complexity of the health consequences of SUB due to dosage-dependent effects, and provides analytical guidance for future research to study the health consequences of behavioural traits.

## Methods
### Comparing different MR methods by simulation and real data analysis
We calibrated 11 commonly used MR methods by simulation, including GSMR2 (implemented in GCTA v1.93.0b, https://yanglab.westlake.edu.cn/software/gcta/index.html#GSMR), IVW, Robust, MR-Egger, weighted median, mode, and Con-Mix, implemented in the R package *MendelianRandomization* (v0.4.2, https://CRAN.R-project.org/package=MendelianRandomization), and MR-Lasso, MR-PRESSO (v1.0, https://github.com/rondolab/MR-PRESSO), MRMix (v0.1.0, https://github.com/gqi/MRMix) and RAPS (v0.2) in R. All the MR methods were used with the default settings. Among these, GSMR2 is an updated version of GSMR[22] and was developed as part of this study (https://github.com/jianyanglab/gsmr2 and https://yanglab.westlake.edu.cn/software/gsmr/). It introduces a new heterogeneity test to exclude invalid IVs and is more robust against directional pleiotropy compared to GSMR (Supplementary Note 1). All the methods were compared based on the false-positive rate, the estimate of causal effect, and statistical power under the scenarios with different proportions of invalid IVs, different proportions of variance explained by the invalid IVs, and different levels of balanced or directional pleiotropy. Detailed simulation settings and results can be found in Supplementary Note 2 and Supplementary Figs. 1–3. We then applied the 11 MR methods to test for causal associations between SUB and common diseases of interest in real data. We selected independent lead SNPs (LD $r^2 < 0.01$ between the lead SNPs) with a GWAS $P$-value $< 5 \times 10^{-8}$ as IVs for the MR analyses. We defined a significant or suggestive association for each exposure using a local FDR of $< 0.01$ or $< 0.05$ (*qvalue* package[23]), respectively. In the bi-directional MR analyses, we also set the $p$-value threshold for selecting IVs for common diseases at $5 \times 10^{-8}$. This p-value threshold is equivalent to a chi-squared statistic of 29.7, but considerably more stringent than the 'rule of thumb' chi-squared statistic threshold of 10 [ref. 24]. Based on the simulation results, we have compiled a table recommending the use of these 11 MR methods for

real data analyses under various circumstances (Supplementary Table 1). Additionally, we conducted a univariate MR analysis for each IV used in GSMR2 for all exposure-outcome pairs and plotted the strength of association of each IV with the exposure (as measured by $p$-value) against its causal estimate ($\hat{b}_{xy}$) (Supplementary Data 1).

### Phenotype definitions and selection criteria
We collected seven traits related to substance use behaviours (Supplementary Table 2) from the UK Biobank (UKB) data[21]. We obtained access to the UK biobank data by applying to the Access Management Team under Application Numbers 12514 and 66982. The UK Biobank has obtained ethical approval from the North West Multi-centre Research Ethics Committee (MREC) as a Research Tissue Bank (RTB), which means that researchers are not required to seek separate ethical approval and can process the data under the existing RTB approval. The smoking status was defined based on the answer to questions about current and past tobacco smoking (data-field IDs: 1239 and 1249). Individuals who answered "just tried once or twice" for past tobacco smoking were regarded as never regular smokers. For smoking initiation (SI), we collected 453,693 records from a self-report survey (208,988 regular smokers and 244,705 never regular smokers) and coded regular smokers as 1 and never regular smokers as 0. Former smoking (FS) was also a binary trait, contrasting between 161,569 former smokers and 244,705 never regular smokers. Binary trait current smoking (CS) was defined to the contrast between 47,419 current smokers and 244,705 never regular smokers. Cigarette per day (CPD) was a quantitative phenotype measured by how many cigarettes were smoked per day for the current smokers (data-field ID: 3456) who mainly smoked manufactured or hand-rolled cigarettes (data-field ID: 3446). Smoking cessation (SC) was a binary trait, contrasting between 161,569 former smokers and 47,419 current smokers, where former smokers were defined as participants who had quit smoking, and current smokers were defined as participants who reported that they were smoking at the time of the interview. For alcohol consumption (AC), we calculated an average intake of alcohol consumption in units per week[25] ($n = 358{,}449$ individuals). We performed a correction for misreports and longitudinal changes, similar to that in our previous study[26] which shows that not all the MR methods are robust to these confounders. Heavy alcohol consumption (HAC) was a binary trait (coded as 1 or 0), defined as current heavy drinkers ($n = 106{,}576$, mean $= 21.23$ units, standard deviation (s.d.) $= 8.54$) who drink $\geq 12.5$ units per week, contrasted to never drinkers. Moderate alcohol consumption (MAC) was a binary trait, defined as current moderate drinkers ($n = 251{,}873$, mean $= 5.74$ units, s.d. $= 3.48$) who drink $< 12.5$ units per week, contrasted to the never drinkers ($n = 14{,}488$). We chose the threshold of 12.5 units per week because it showed the lowest risk of all-cause mortality in a previous study[10]. For coffee intake (CI), the number of cups of coffee intake per day (mean $= 2.07$ cups per day, s.d. $= 2.10$) was collected from 421,947 individuals (data-field ID: 1498). For tea intake (TI), the number of cups of tea intake per day (mean $= 3.47$ cups per day, s.d. $= 2.90$) was collected from 440,094 individuals (data-field ID: 1488). Moderate/heavy coffee/tea intake were defined as drinkers consuming $\geq$ or $< 5$ cups per day, contrasted to the non-drinkers. Diet by 24 h recall is an online-follow questionnaire being emailed to participants at 3–4 monthly intervals (category ID: 100090). The phenotype "coffee consumed" (data-field ID: 100240) is a binary trait indicating whether coffee intake in the last 24 h ($n = 63{,}891$). Sugar and artificial sweetener added to the coffee (data-field ID: 100370 and 100380) are quantitative traits measured by the number of teaspoons per drink, with half, 1, 2, and 3+ coded as 0.5, 1, 2, 3, respectively ($n = 63{,}786$; those answered "varied" were excluded).

The phenotypic records of 18 common diseases in the UKB were acquired from ICD10 main diagnoses, ICD10 secondary diagnoses, and self-report records (data-field IDs: 41202, 41204, and 20002; $n = 454{,}108$-455,607). We first selected the same 22 common diseases as in ref. 22 but excluded 4 diseases with a low prevalence ( $< 2\%$ in UKB v2 full release). Each disease trait was labelled as 0 (control) or 1 (disease carrier), and the disease count was the number of diseases carried by an individual as an

indicator to quantify the general health status of the UKB participants. The descriptive characteristics of these phenotypes can be found in Supplementary Table 3. We also collected two socioeconomic traits, educational attainment (EA) and household income (HI), from the UKB. EA was measured by years of schooling derived from qualification (data-field ID: 6138), and HI was measured by annual average total household income before tax (data-field ID: 738).

Considering the concerns that not all the methods we used are free from bias due to sample overlap, as some disease summary statistics also incorporate the UKB data, we performed a re-sampling analysis to compare the estimates of causal effect and their corresponding test-statistics (i.e., z-scores) between the analyses with and without sample overlap, given the same sample size. To do this, we randomly divided the UKB participants into two equal subgroups and re-ran the GWAS and MR for smoking initiation and cardiovascular disease. We repeated this process 100 times and compared the $b_{xy}$ estimates between the scenarios of no sample overlap and full overlap for each method. We did not observe any significant difference in the $b_{xy}$ estimate or z-test statistic between the analyses with no and full sample overlap, except for Egger, which presented a significantly higher $b_{xy}$ estimate and test-statistic in full overlap compared to no overlap (Supplementary Fig. 4). It is noteworthy that inflation in test-statistics due to sample overlap is a well-recognized issue in two-sample MR methods, and while the simulations in this study suggest that the primary conclusions are highly unlikely to be influenced by sample overlap, they should not be misconstrued as dismissing the issue of sample overlap entirely.

### GWAS and genetic correlation

The UKB individual-level genotype data were subject to quality controlled and imputed to Haplotype Reference Consortium (HRC)[27] by the UKB data analysis team[21]. We extracted a subset of the UKB data representing European ancestry ($n = 456,426$) by projecting all the participants onto the principal components (PCs) from the 1000 Genomes Project (1KGP). Then, we used PLINK2[28] (https://www.cog-genomics.org/plink2) to generate the hard-call genotypes from the imputed genotype probabilities (parameter setting: -hard-call 0.1). We filtered out SNPs with minor allele count < 5, missing genotype rate > 0.05, Hardy-Weinberg equilibrium test $P$-value $< 1 \times 10^{-6}$, or imputation info score < 0.3.

We used BOLT-LMM[29] to perform GWAS to acquire summary statistics for SUB and common diseases in the UKB. For binary traits (case versus control), we ran BOLT-LMM analysis fitting sex, age, and first 10 PCs as covariates, and then transformed the effect size from BOLT-LMM effects to the odds ratio (OR) using LMOR[30]. For quantitative traits (e.g., SUB and disease count), we excluded the extreme phenotypic values located outside the mean $\pm 7$ s.d. interval in each sex group, pre-adjusted the phenotypes for sex and age, converted them to z-scores, and then performed BOLT-LMM analysis[29] with the first 10 PCs fitted as covariates. Recently developed approaches[31–33] have been utilized to perform generalized linear mixed model-based association analysis for binary traits in biobank-scale datasets. We employed fastGWA-GLMM[33] to rerun the GWAS and subsequent MR analyses for the four smoking related binary traits. The effect sizes of genome-wide significant SNPs were highly similar between GLMM and LMM + transformation (e.g., a Pearson's correlation of 0.9996 for 157 independent SNPs for SI). The causal estimates were also largely consistent, and any discrepancy was mainly due to the low robustness of certain MR methods rather than the methods used to generate GWAS summary statistics (Supplementary Fig. 5). GWAS summary statistics for several common diseases were obtained from the published studies: coronary artery disease (CAD)[34], type 2 diabetes (T2D)[35], Crohn's disease (CD)[36], ulcerative colitis (UC)[36], rheumatoid arthritis (RA)[37], schizophrenia (SCZ)[38], bipolar disorder (BIP)[39], major depressive disorder (MDD)[40], Alzheimer's disease (AD)[41], ovarian cancer[42], breast cancer[43], and prostate cancer[44]. The descriptive characteristics of these phenotypes can be found in Supplementary Table 4.

Genetic correlation characterizes the genetic relationship between two traits due to pleiotropic and/or causality. To estimate the genetic correlation

between substance use behaviours, we used bivariate LDSC[45] which only requires GWAS summary statistics. The input for bivariate LDSC was restricted to ~1.2 million SNPs that overlapped with those in the HapMap 3 panel.

### Multi-trait-based conditional and joint analysis (mtCOJO)

The mtCOJO method[22] (https://yanglab.westlake.edu.cn/software/gcta/index.html#mtCOJO) is an approach to conduct GWAS for a trait, conditioned on a set of other traits, using only summary statistics. To validate the results from the mtCOJO analysis, we ran the BOLT-LMM analysis for the seven main SUB traits with EA and HI fitted as covariates in the linear mixed model. Then, we used the conditional GWAS summary to perform the MR analysis and compared the results with the unconditional results (Supplementary Fig. 6).

### Investigating dose-dependent effects

We conducted a simulation to analyse the causal relationship between exposure ($x$) and outcome ($y$). Specifically, we simulated a quadratic relationship between $x$ and $y$ ($y = x^2 + x$), divided $x$ into ten quantiles based on exposure values, and classified the first quantile as the control group (i.e., those who never drink coffee or tea). The causal effect ($b_{xy}$) is set as 0.2. We also identified moderate and heavy intake groups based on the turning point of the average outcome value (i.e., disease risk). We then conducted GWAS of the moderate and heavy intake groups against the control group and estimated the genetic correlation ($r_g$) between $x$ and $y$ in each group. We repeated this simulation 100 times for both linear and non-linear causal effects and then compared the estimates (Supplementary Fig. 7).

There are concerns that dichotomizing consumption data does not provide direct evidence for dose-dependent effects. To address this concern, we used a recently developed method called PolyMR[46] to investigate non-linear causal effects (Supplementary Fig. 8). However, since PolyMR was not designed for binary outcomes, we selected six quantitative biomarkers (total cholesterol, blood glucose, HbA1c, HDL, LDL, triglycerides, and urate) as outcomes to assess the potential non-linear causal effects of coffee intake and tea intake (CI and TI).

### Reporting summary

Further information on research design is available in the Nature Portfolio Reporting Summary linked to this article.

## Results

### Comparison of the commonly used MR methods

Prior research[26] has indicated that some GWAS on substance use behaviours (SUBs) may be biased by potential confounders, leading to invalid IVs. This necessitates the re-evaluation of MR methods through simulation. To compare the performance of different MR methods and better understand the differences in real data analysis results, we conducted extensive simulations under a range of scenarios (Methods and Supplementary Note 2), with a specific focus on the effect of horizontal pleiotropy. In our previous study[26], we observed that strong confounders could distort the true genetic correlation and causal association, even reversing their direction, and a large proportion of IVs showed strong directional pleiotropic effects. Thus, simulation settings need to mimic such extreme scenarios to test the limits of MR methods. We included in the benchmark analysis a set of commonly used MR methods, namely IVW[47], weighted median[48], mode[19], MR-Egger[18], Robust[49], MR-Lasso[50], RAPS[51], MR-PRESSO[20], MRMix[52] and Con-Mix[53]. We also included an upgraded version of the Generalised Summary-data-based Mendelian Randomisation[22] (named GSMR2), incorporating a new global heterogeneity test with improved robustness to detect and remove pleiotropic IVs (Supplementary Note 1).

The simulation results showed that under the null model (i.e., no causal effect between exposure and outcome), when the proportion of invalid IVs was small and the invalid IVs explained a small fraction of heritability of the exposure, almost all the methods showed a well-controlled false-positive rate (FPR) (Supplementary Figs. 1, 2). However, when the proportion of

invalid IVs was large (e.g., half of the IVs were invalid), most methods had an inflated FPR under the null. Under a causal model (i.e., the alternative model), the estimates of causal effects could be biased by directional pleiotropy (the effects of pleiotropic IVs are correlated between exposure and outcome), and the inflation was proportional to the strength of the directional pleiotropy. Under the alternative model, most methods attained high statistical power when the level of directional pleiotropy was modest, but the power for several methods decreased substantially when the level of directional pleiotropy was strong (Supplementary Fig. 3). The simulation results suggest that in the presence of strong directional pleiotropy, no MR method can attain both low FPR under the null model and high statistical power under a causal model. The consistency in result between the MR methods reduced with the increased level of directional pleiotropy because of the differences in robustness to pleiotropy between the methods. Thus, it is essential to compare results from different MR methods before making definitive inferences about causality, and such triangulation framework has proven effective in improving the robustness of causal inference[50,54,55].

To estimate the causal effects of SUB on common diseases, we carried out MR analyses between 7 SUB traits (e.g., smoking initiation, alcohol consumption) and 18 common diseases (e.g., asthma, type 2 diabetes, psychiatric disorders) plus disease count (i.e., a sum of diseases carried) using the 11 MR methods that were calibrated as mentioned above (Methods and Supplementary Tables 2, 3). For each SUB trait, we used a local false discovery rate (FDR) of 0.01 to define significant associations between the exposures and outcomes and a local FDR of 0.05 to define suggestive associations (Methods).

## Widespread risk effects of smoking on common diseases

Results from nearly all the methods consistently showed that smoking initiation (SI) had significant risk effects on 13 diseases and protective effects against 1 disease (Fig. 1 and Supplementary Data 2), consistent with recent MR studies for smoking traits[56–58]. In total, there were 100 significant associations out of 209 tests (11 methods multiplied by 19 outcomes). MR-Egger and MRMix were the only two methods that did not show any significant association, consistent with the simulation evidence that they had the lowest statistical power in most scenarios among the MR methods tested (Supplementary Fig. 3). The only protective effect of SI was against allergic rhinitis, which was significant in seven methods, and the estimates were largely consistent across methods (Fig. 1). Former smoking (FS) and current smoking (CS) both showed consistent results with SI, and all these three smoking-related traits showed consistent risk effects on disease count (Supplementary Figs. 9 and Supplementary Data 3, 4). On the contrary, for smoking cessation (SC), only the GSMR2 method showed a significant protective effect against cardiovascular disease and a suggestive protective effect against disease count (Supplementary Fig. 9 and Supplementary Data 5). The small number of significant associations for SC was likely due to the lack of power because only 8 index SNPs were included in the MR analysis (see below for the results from a more powerful analysis).

## The observed beneficial health effects of moderate drinking are likely non-causal

The health consequences of alcohol consumption (AC) have been under debate for decades. Several genetic analyses showed negative estimates of

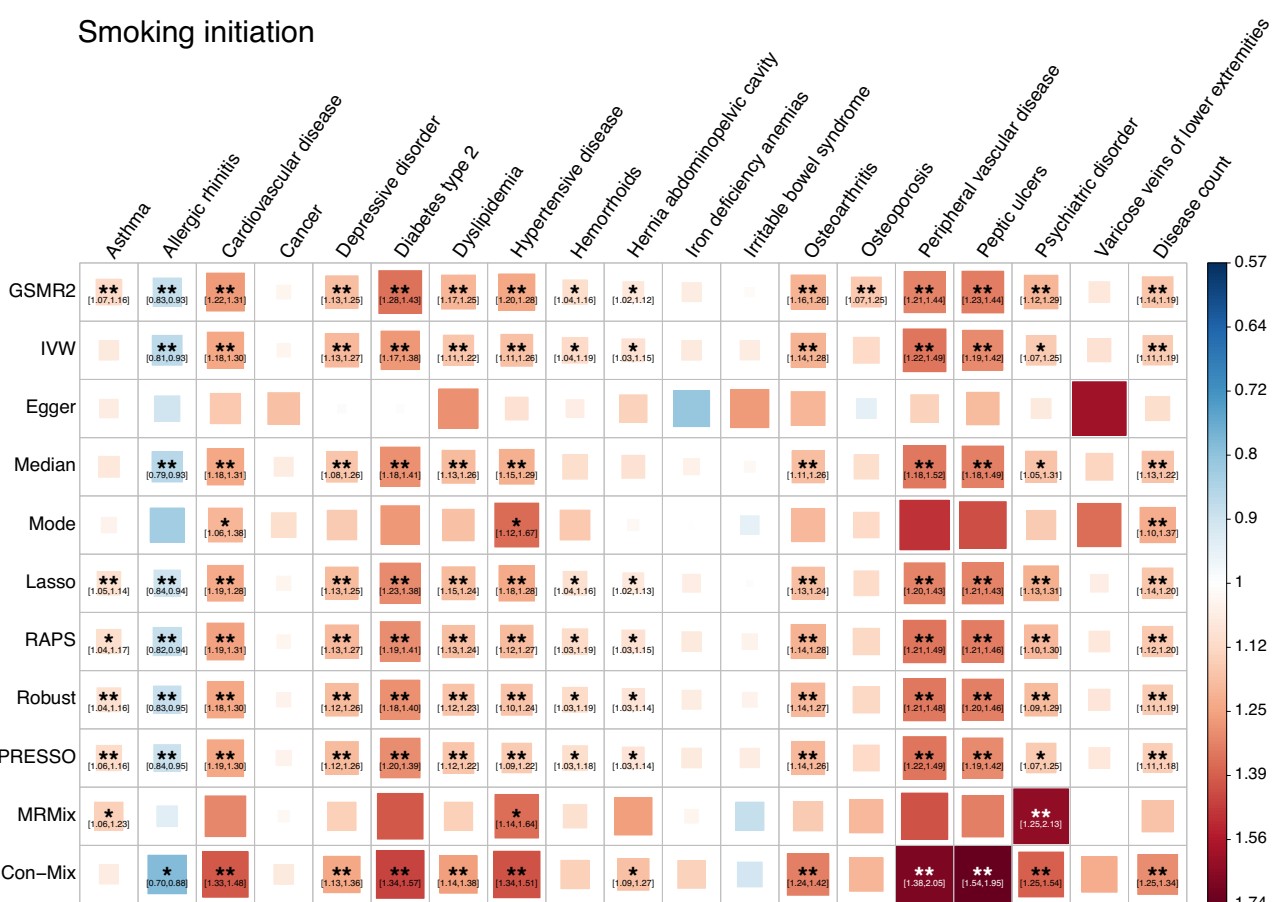

**Fig. 1 | Estimates of causal associations between smoking initiation and common diseases from different MR methods.** Data for both the exposures and the outcomes were from the UK Biobank. In total, 152 index SNPs for smoking initiation were used as instrumental variables. Each square represents an MR association result, and the colour indicates the direction of the estimate. The size and opacity of each square is proportional to the size of causal effect estimate. The suggestive associations (local FDR < 0.05) are annotated with "*" and "[95% confidence interval of OR]". The significant estimates with local FDR < 0.01 are labelled with "**". The names of the MR methods are shown on the left of the plot, and the names of the diseases are shown on the top.

genetic correlation between AC and common diseases[25,59]. However, recent MR analyses failed to find any significant cardioprotective effect of alcohol drinking[58,60,61]. In addition, observational studies showed a non-linear relationship of AC with common diseases[10,62], e.g., a J-shaped relationship with cardiovascular disease[63,64]. Our previous study showed that the negative estimates of genetic correlation and J-shaped relationship between AC and disease could be largely driven by misreports and longitudinal changes due to disease ascertainment[26]. In this study, results from different MR methods showed consistently that AC had risk effects on cardiovascular disease, dyslipidemia, and hypertensive disease (Fig. 2 and Supplementary Data 6). To further investigate the health effects of moderate drinking, we derived two additional phenotypes: moderate alcohol consumption (MAC) and heavy alcohol consumption (HAC) and re-ran the MR analysis (Methods). We found that MAC did not show any significant protective effects in any methods, while HAC still showed significant risk effects on dyslipidemia and hypertensive disease (Supplementary Fig. 10 and Supplementary Data 7), implying that the protective effects of moderate drinking observed from observational studies are likely non-causal.

### Coffee and tea intake exerted complicated effects on common diseases

Coffee intake (CI) showed significant risk effects on five diseases (Fig. 3). For asthma, cardiovascular disease, dyslipidemia, and iron deficiency anemias, only one method was significant although the estimates from all the other methods showed a consistent direction. For osteoarthritis, all the methods showed significant results except for MR-Egger, and the direction of the

estimates were all consistent (Supplementary Data 8 and Fig. 3). The mean OR from all methods was 1.52, which is interpreted as a 1 s.d. increase in CI (equivalent to 2.10 cups per day) leading to a 1.52-fold increase in the risk of osteoarthritis. CI also showed a protective effect against irritable bowel syndrome, osteoporosis, and varicose veins of lower extremities (VVLE), but the evidence is considered as modest since only a few methods provided significant estimates (Fig. 3). Tea intake (TI) showed a significant risk effect on osteoarthritis in Median and Mode methods, a significant protective effect against osteoporosis in GSMR2, and a suggestive protective effect against type 2 diabetes (T2D) and VVLE in GSMR2 and Mode methods (Fig. 4), indicating possible confounding effects were dealt with differently by different methods so that these results should be interpreted with great caution. Neither CI nor TI had a significant effect on disease count, suggesting that the overall health effects of these two behaviours are mild (Figs. 3, 4 and Supplementary Data 8, 9). Alternatively, the effect may be dosage-dependent, and thus underestimated if we assume a linear relationship (see below).

The relationship between CI and common diseases is complicated and controversial. For example, CI has previously been associated with lower T2D risk[65,66]. However, recent evidence argues that high coffee consumption increases the T2D risk compared to low consumption[67]. We attempted to investigate this potential dosage-dependent relationship via a stratified analysis by performing logistic regression of 18 common diseases on 10 different dose groups against non-drinkers (Supplementary Note 3). The results showed that for T2D, coffee intake of less than five cups per day had beneficial effects, but when the intake was more than six cups per day, the

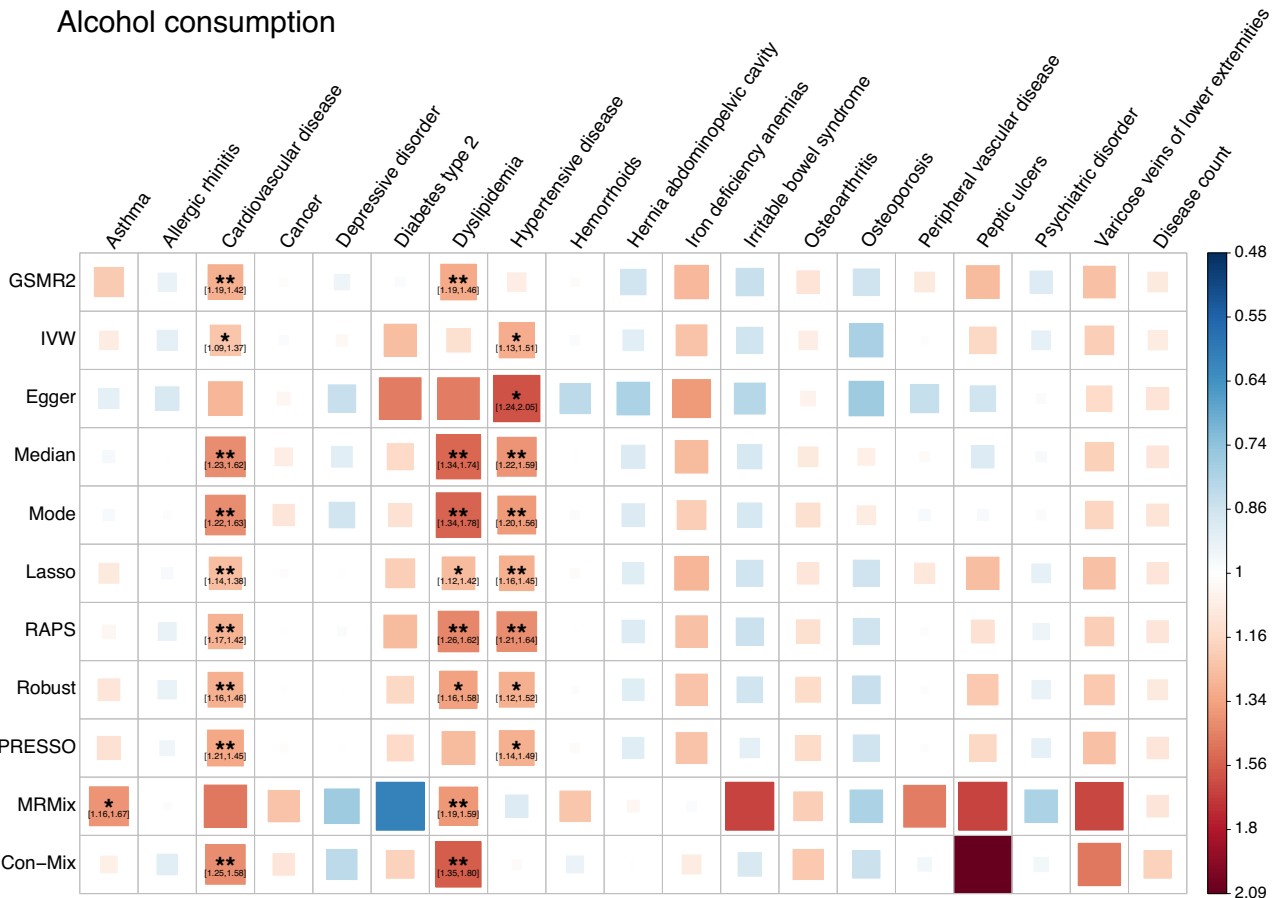

**Fig. 2 | Estimates of the causal associations between alcohol consumption and common diseases from different MR methods.** Data for both the exposures and the outcomes were from the UK Biobank. In total, 52 index SNPs for alcohol consumption were used as instrumental variables. Each square represents an MR association result and the colour indicates the direction of the estimate. The size and opacity of each square is proportional to the size of causal effect estimate. The suggestive associations (local FDR < 0.05) are annotated with "*" and "[95% confidence interval of OR]". The significant estimates with local FDR < 0.01 are labelled with "**". The names of the MR methods are shown on the left of the plot, and the names of the diseases are shown on the top.

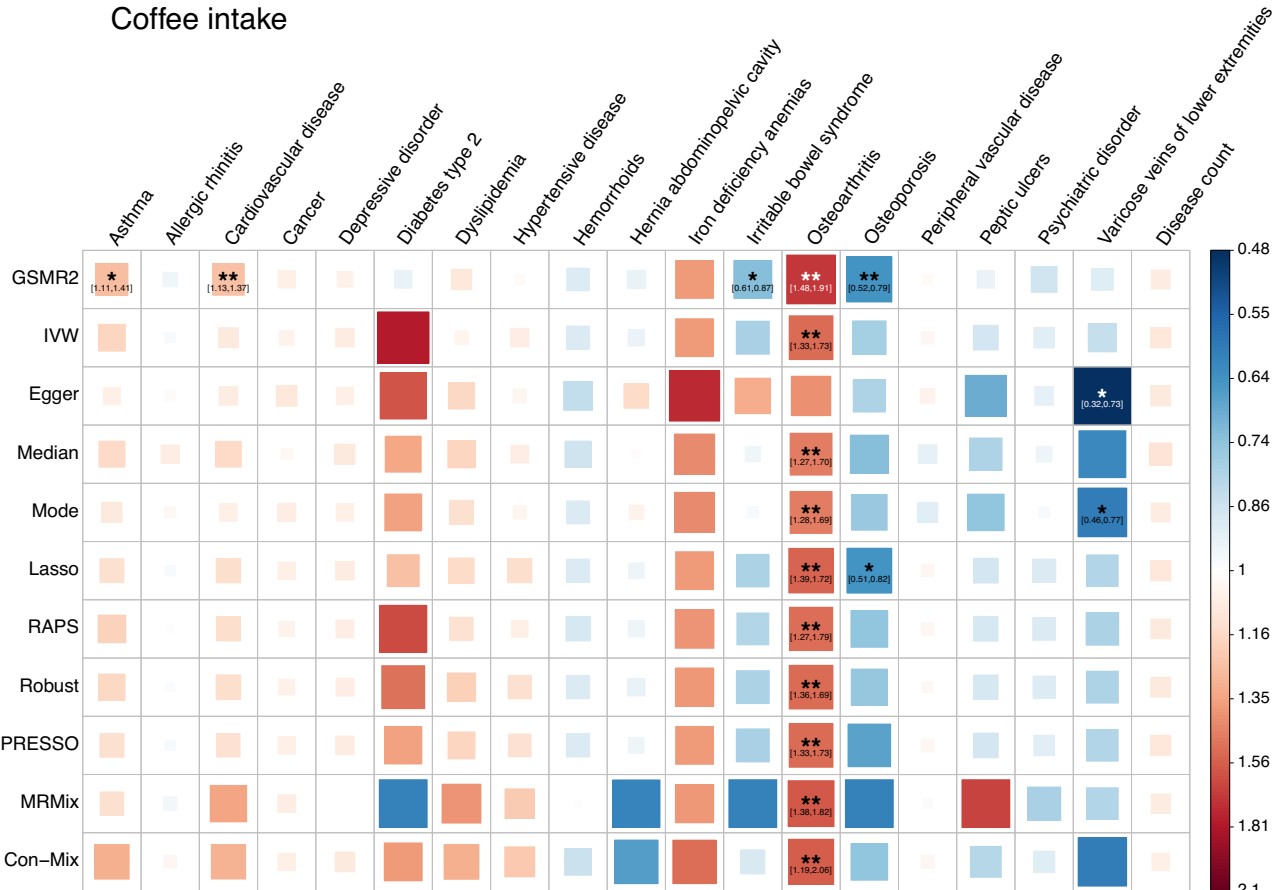

**Fig. 3 | Estimates of the causal association between coffee intake and common diseases from different MR methods.** Data for both the exposures and the outcomes were from the UK Biobank. In total, 49 index SNPs for coffee intake were used as instrumental variables. Each square represents an MR association result, and the colour indicates the direction of the estimate. The size and opacity of each square is proportional to the size of causal effect estimate. The suggestive associations (local FDR < 0.05) are annotated with "*" and "[95% confidence interval of OR]". The significant estimates with local FDR < 0.01 are labelled with "**". The names of the MR methods are shown on the left of the plot, and the names of the diseases are shown on the top.

protective effects turned to risk effects (Supplementary Fig. 11). TI also showed dosage-dependent patterns for cardiovascular disease and osteoarthritis (Supplementary Fig. 12) as well as several other diseases, suggesting that the health effects of both coffee and tea intake might be dosage-dependent.

If there is a J-shaped, dosage-dependent relationship between CI/TI and a disease, the genetic correlation ($r_g$) between moderate CI/TI and the disease could potentially be in the opposite direction compared to that between high CI/TI and the disease. To verify this hypothesis, we derived four new traits, heavy/moderate coffee intake (HCI/MCI) and heavy/moderate tea intake (HTI/MTI), i.e., contrasting people with a daily intake of ≥ 5 or < 5 cups against those with zero intake, and assessed the associations of the original and new tea/coffee intake phenotypes with the 18 common diseases by genetic correlation, stratified regression, and MR analyses (Methods). HCI showed a significant (local FDR < 0.01) positive $\hat{r}_g$ with 3 diseases and no significant negative $r_g$ (Supplementary Fig. 13), consistent with the results for CI. In contrast, MCI showed a significant negative $\hat{r}_g$ with 9 diseases and no significant positive $\hat{r}_g$ (Supplementary Fig. 13 and Supplementary Data 10). For example, MCI showed a negative $\hat{r}_g$ (−0.22, s.e. = 0.03, q − value = $2.65 \times 10^{-10}$) with cardiovascular disease, whereas the estimate for HCI was in the opposite direction ($\hat{r}_g = 0.16$, $s.e. = 0.04$, q − value = $1.07 \times 10^{-4}$), consistent with the results from the dosage-dependent regression analysis (Supplementary Fig. 11). However, in the MR analysis, the significant estimates of causal effects ($\hat{b}_{xy}$) of MCI on common diseases were mostly in consistent direction

with those for HCI (Supplementary Fig. 14), suggesting that the difference in the direction of $\hat{r}_g$ with common diseases between MCI and HCI might be due to pleiotropic effects and/or confounders (see below for more discussion). For tea intake, the $r_g$ estimates between the MTI-disease pairs were broadly consistent with those between the HTI-disease pairs, e.g., both MTI and HTI showed significant negative genetic correlation with T2D (Supplementary Fig. 13) and protective effects against T2D as suggested by three MR methods (Supplementary Fig. 15). The only robust risk causal effect of HTI was found for osteoarthritis (significant in MR-Median and MR-Mode methods with the estimates from the other MR methods in a consistent direction). We further demonstrated by simulation that observing opposing directions in the estimates of $r_g$ between exposure and outcome across different stratified exposure groups is indicative of dosage-dependent effects (Methods and Supplementary Fig. 7).

To better understand the dosage-dependent associations and the discrepancies between the genetic correlation and MR results shown above, we focused specifically on the association between MTI/HTI and osteoarthritis because of a discernible dosage-dependent effect shown consistently in the stratified regression, genetic correlation, and MR analyses (Supplementary Figs. 12, 13, 15). We visualized the relationship between the effects of IVs on the exposure and outcome (Supplementary Fig. 16) and found that the two most significant IVs (rs1264377 and rs977474) for MTI were distinct from those for HTI (rs4410790 and rs2472297), causing the estimates of $b_{xy}$ to be in opposite directions for the two sub-phenotypes (Table 1). For example, the T allele of rs4410790 (top IV for HTI) had a negative effect on HTI

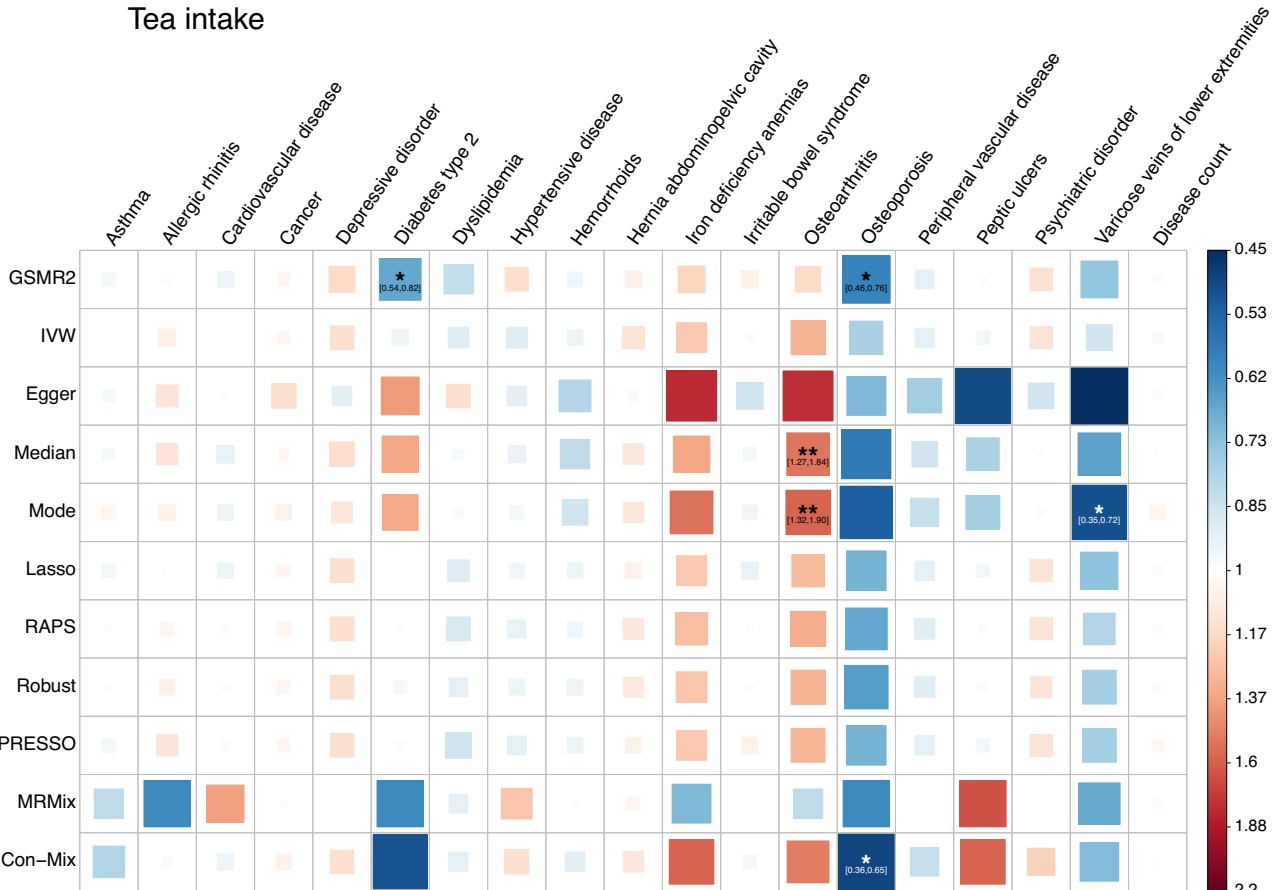

**Fig. 4 | Estimates of the causal association between tea intake and common diseases from different MR methods.** Data for both the exposures and the outcomes were from the UK Biobank. In total, 43 index SNPs for tea intake were used as instrumental variables. Each square represents an MR association result, and the colour indicates the direction of the estimate. The size and opacity of each square is proportional to the size of causal effect estimate. The suggestive associations (local FDR < 0.05) are annotated with "*" and "[95% confidence interval of OR]". The significant estimates with local FDR < 0.01 are labelled with "**". The names of the MR methods are shown on the left of the plot, and the names of the diseases are shown on the top.

$(\hat{b}_{GWAS} = -0.091, s.e. = 0.007, P = 4.90 \times 10^{-35})$ but a positive effect on MTI $(\hat{b}_{GWAS} = 0.034, s.e. = 0.006, P = 3.40 \times 10^{-8})$. All these observations above indicate a substantial genetic heterogeneity between MTI and HTI, which was further supported by the evidence that the genetic correlation between them was significantly different from unity $(\hat{r}_g = 0.651, s.e. = 0.028)$. For coffee intake, the top two IVs for HCI and MCI were the same, and their $b_{xy}$ estimates were in the same direction (Table 1). Also, HCI showed more significant $b_{xy}$ estimates with common diseases (risk effects on T2D, dyslipidemia, and osteoarthritis, and protective effects against osteoporosis across different methods) than MCI did (only risk effect on T2D and osteoarthritis in a single method, Supplementary Fig. 14). In addition to the disease outcomes, we utilised PolyMR to directly assess the non-linearity of the effects of CI and TI on seven common biomarkers (Methods). We found significant non-linear effects of CI on total cholesterol $(P = 9.50 \times 10^{-7})$ and low-density lipoprotein levels $(P = 2.46 \times 10^{-3})$, even after applying the Bonferroni correction (Supplementary Fig. 8).

Taken together, our results demonstrate the complexity of the health consequences of coffee/tea intake. The results also suggest that the overall health effects of CI and TI are mild and need to be interpreted with caution, especially when a dosage-dependent relationship is observed.

**Validating the causal estimates using data from published studies**

To validate our causal estimates above, we first re-ran the MR analysis with the disease GWAS summary statistics replaced by those from published studies (Methods and Supplementary Table 4). We identified 86 significant associations (local FDR < 0.01) between the 7 SUB traits and 12 common diseases (Supplementary Fig. 17). The causal effects estimated using the published disease GWAS data were highly correlated with those estimated using the UKB disease data, despite the phenotypic definitions of the diseases being slightly different between studies. The Pearson's correlation $r$ of the $b_{xy}$ estimates across 7 SUB traits was 0.86 between cardiovascular disease and coronary artery disease (CAD), and 0.77 between psychiatric disorder and schizophrenia (SCZ) (note: the reported $r$ is the median of the estimates across 11 MR methods). We also re-ran the MR analysis using summary data for SI, SC, and AC from a recent GWAS meta-analysis by the GSCAN consortium[59] (Supplementary Fig. 18). The $b_{xy}$ estimates using the UKB SUB data were generally consistent with those using the GSCAN SUB data (Pearson's correlation $r = 0.55–0.81$ across different MR methods). Of the 100 significant associations between SI and common diseases discovered in the UKB, 94 remained significant when using SI from the GSCAN data. Notably, smoking cessation from GSCAN showed several significant protective effects with consistent estimates from multiple methods, validating the beneficial effects of SC, as indicated by the GSMR2 analysis above with the SC data from the UKB. The gain of power is likely due to the increased number of IVs (from 8 to 18). These results also demonstrate the power of GSMR2 when the number of IVs is limited. On the other hand, the replication rate of AC was low (4/20), probably because the GSCAN dataset has not corrected for misreports and longitudinal changes as noted previously[26].

**Table 1 | Dosage-dependent effects of the top four GWAS signals for coffee and tea intake**

| SNP | Nearest Gene(s) | A1/A2 | Trait | N | Freq_A1 | beta | s.e. | *P* |
|---|---|---|---|---|---|---|---|---|
| rs4410790 | *AHR* | T/C | CI | 421947 | 0.366 | −0.062 | 0.002 | 1.1E-166 |
| | | | MCI | 393101 | 0.369 | −0.067 | 0.005 | 5.0E-36 |
| | | | HCI | 124890 | 0.369 | −0.200 | 0.010 | 5.3E-92 |
| | | | TI | 440094 | 0.367 | −0.045 | 0.002 | 4.1E-92 |
| | | | MTI | 351508 | 0.373 | 0.034 | 0.006 | 3.4E-08 |
| | | | HTI | 155936 | 0.354 | −0.091 | 0.007 | 4.9E-35 |
| rs2472297 | *CYP1A1* | T/C | CI | 421947 | 0.265 | 0.074 | 0.002 | 2.6E-198 |
| | | | MCI | 393101 | 0.263 | 0.080 | 0.006 | 1.2E-40 |
| | | | HCI | 124890 | 0.264 | 0.250 | 0.010 | 4.8E-130 |
| | | | TI | 440094 | 0.264 | 0.060 | 0.002 | 1.9E-134 |
| | | | MTI | 351508 | 0.258 | −0.027 | 0.007 | 9.2E-05 |
| | | | HTI | 155936 | 0.278 | 0.124 | 0.008 | 1.3E-54 |
| rs1264377 | *PSORS1C3, MIR877* | A/G | CI | 421947 | 0.182 | 0.015 | 0.0028 | 4.0E-08 |
| | | | MCI | 393101 | 0.181 | −0.004 | 0.0067 | 0.51 |
| | | | HCI | 124890 | 0.185 | 0.043 | 0.0119 | 3.5E-04 |
| | | | TI | 440094 | 0.182 | −0.002 | 0.0028 | 0.37 |
| | | | MTI | 351508 | 0.181 | −0.054 | 0.0077 | 2.8E-12 |
| | | | HTI | 155936 | 0.185 | −0.028 | 0.0091 | 1.8E-03 |
| rs977474 | *PRH1/ PPR4/TAS2R14* | C/T | CI | 419168 | 0.166 | 0.022 | 0.0029 | 1.8E-13 |
| | | | MCI | 390497 | 0.165 | 0.044 | 0.0071 | 4.6E-10 |
| | | | HCI | 124071 | 0.163 | 0.077 | 0.0125 | 7.1E-10 |
| | | | TI | 437186 | 0.165 | −0.021 | 0.0029 | 4.0E-13 |
| | | | MTI | 349186 | 0.166 | −0.058 | 0.0080 | 6.3E-13 |
| | | | HTI | 154901 | 0.167 | −0.067 | 0.0095 | 1.2E-12 |

rs4410790 and rs2472297 are the top two GWAS signals in all GWAS except for MTI, and rs1264377 and rs977474 are the top two GWAS signals in MTI GWAS. A1/A2: minor allele/major allele. *Freq_A1* allele frequency of A1, *N* sample size, *beta* estimate of SNP effect, *s.e.* Standard error of the beta estimate.

## Causal estimates are largely robust to the confounding of socioeconomic status

Considering that the estimates of causal associations between SUB and common diseases might be confounded by SES, we estimated the causal effects of SUB on the diseases adjusting for educational attainment (EA) and household income (HI). To achieve this, we applied mtCOJO[22] which only requires summary statistics to conduct a conditional GWAS analysis for each SUB or disease trait conditioning on EA and HI simultaneously (Methods). We then re-ran the MR analysis using the SES-adjusted SUB and disease GWAS summary statistics. The causal estimates after the SES adjustment were largely consistent with those without adjustment (Fig. 5 and Supplementary Data 11), indicating that the causal estimates between SUB and common diseases were generally robust to the confounding of the SES analyzed in this study (except for MRMix which showed several extreme $b_{xy}$ estimates for AC and TI after the SES adjustment). In terms of the robustness for each specific exposure, most of them showed consistent results before and after the SES adjustment except for tea/coffee intake. The results for smoking-related traits were highly robust even for the results from MRMix. These observations indicate that tea/coffee intake is more likely to be confounded by SES compared to smoking and drinking. We further validated the mtCOJO adjustment by conducting BOLT-LMM[29] analysis on both SUB and common diseases fitting EA and HI as covariates and re-ran the MR analysis (Methods). The individual-level data-based conditional GWAS analysis results were consistent with those from mtCOJO (Supplementary Fig. 6), and the Pearson's correlation *r* of the $b_{xy}$ estimates between mtCOJO adjustment and individual-level data-based adjustment ranged from 0.61 to 0.97 across different exposures (excluding the estimates from MRMix). We also adjusted the SUB and disease GWAS data for two physical activity traits (i.e., leisure screen time and moderate-to-

vigorous intensity physical activity during leisure time)[68] using mtCOJO, and the causal estimates remain largely unchanged (Supplementary Fig. 19). To further assess the robustness of our analysis to potential collider bias[69], we employed Slope-Hunter[70] to correct the seven SUB traits for SES, specifically educational attainment, and re-estimated their effects on diseases using all 11 MR methods. The $b_{xy}$ estimates after Slope-Hunter adjustment were largely consistent with those after mtCOJO adjustment (Supplementary Fig. 20).

## Bi-directional effects are rare

To investigate whether there are reverse causal associations between SUB and complex diseases (i.e., disease status leads to behavioural change), we performed a reverse MR analysis, designating a disease as the exposure and an SUB as the outcome (Methods). The number of diseases that showed significant effects on SUB at local FDR < 0.01 was small (Supplementary Data 12). For the diseases available from the UKB, only asthma showed significant negative effects on current smoking in the GSMR2 and Lasso analyses. For the diseases available from the published studies, there was a strong positive effect of major depressive disorder (MDD) on smoking initiation ($\hat{b}_{xy} = 0.19 \sim 0.28$) significant in 9 out of the 11 MR analyses, consistent with the previous findings[71,72]. Schizophrenia also showed a positive effect on smoking in multiple MR analyses, but the effect size was much smaller than that for MDD (Supplementary Data 12).

## Discussion

In this study, we investigated the causal associations between substance use behaviours and common diseases. The results showed that SUB typically had detrimental effects on health, irrespective of socioeconomic status.

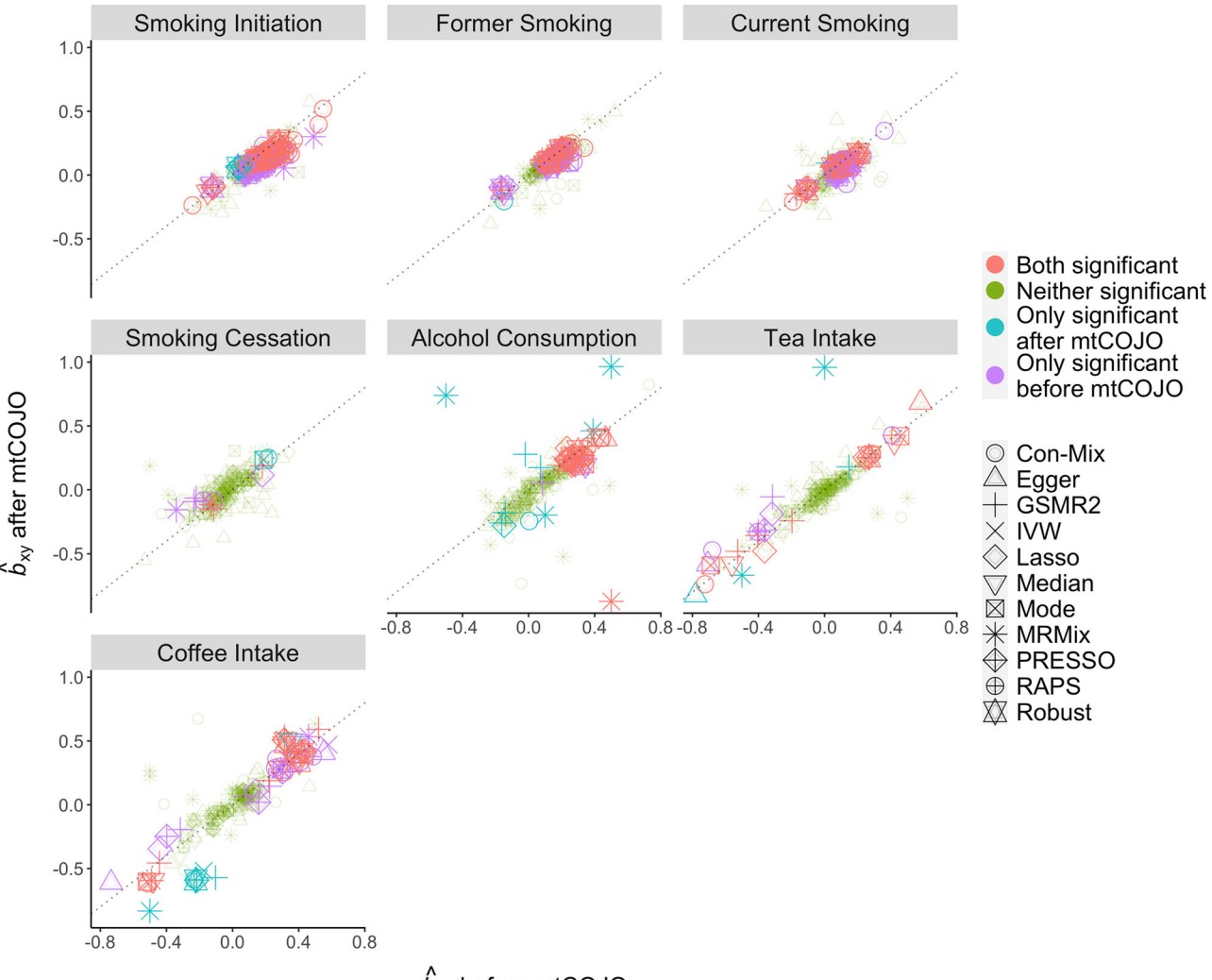

**Fig. 5 | Comparison of the estimates of causal effects of substance use behaviours on common diseases before and after adjusting for socioeconomic status.** This figure shows the comparison of $b_{xy}$ estimates before and after adjusting for SES. Each panel indicates the exposure used. The color of each dot indicates the significance level change before and after SES adjustment. Significant level is defined at local FDR < 0.01. The shape of each dot indicates the MR method used. The grey dashed line is the diagonal line of the coordinate plane.

While smoking behaviours either at present or in the past increased the risk of nearly all common diseases, our results suggested that smoking cessation had beneficial effects on several diseases such as cardiovascular disease, dyslipidemia, and hypertensive disease. We also showed that no significant protective effects were detected for alcohol consumption, including moderate alcohol consumption. Moreover, coffee and tea intake showed complicated relationships with common diseases, and their overall health effects were mild. The effects seemed to be dosage-dependent, and the pattern of dosage-dependence seemed disease-specific.

Among all the tests for smoking, only allergic rhinitis was found with a significant negative association, and such an effect remains debated in the literature. For example, Eriksson et al. [73] showed that smoking was associated with a low prevalence of allergic rhinitis in men, whereas ref. 74 meta-analysed 97 studies and concluded that active smoking was not associated with allergic rhinitis, but passive smoking was. There could be multiple reasons for the inconsistent observations. First, there are different smoking measurements such as smoking initiation (SI) and smoking intensity (measured by cigarettes per day, i.e., CPD). There is genetic heterogeneity between different smoking phenotypes as reported previously[75,76] and observed in this study. For example, the top GWAS signal for SI (rs9919670,

$P = 1.5 \times 10^{-49}$) was not genome-wide significant in CPD GWAS ($P = 0.0079$). Similarly, the top signal in CPD GWAS (rs146009840, $P = 1.2 \times 10^{-52}$) was not genome-wide significant in SI GWAS ($P = 5.1 \times 10^{-5}$). Such differences could lead to different causal estimates from MR. Second, there are differences in the definitions of cases and controls between studies, especially if cases include self-report individuals[77]. Hence, the putative causal association between smoking and allergic rhinitis warrants replication with independent datasets in the future. Third, as pointed out by Saulyte et al. [74], passive smoking had a risk effect on allergic rhinitis, whereas our study only included active smoking and thus the effect of passive smoking was not considered.

Our analysis revealed the complicated effects of coffee and tea intake on common diseases. However, the underlying biological mechanisms are still unclear. Several previous studies have shown that the sugar/sweetener added along with these drinks could confound the associations[78,79], which might be one of the reasons why CI/TI exerted complicated effects on common diseases. In other words, the correlation between CI/TI and metabolic diseases could be confounded by the added sugar/sweetener. According to a 24 h diet recall in the UKB data, around 30.3% of participants added sugar/sweetener into their coffee (data-field ID: 100240, $n = 45,068$),

suggesting that adding sugar/sweetener is common for the coffee drinkers so that such an effect should not be neglected. Unfortunately, these records are not matched with the general coffee and tea intake data in the UKB so that they cannot be directly used as covariates for adjustment, and this issue also applies to data for added milk in coffee or tea. We showed a significant level of genetic heterogeneity between general CI and CI from 24 h diet recall ($\hat{r}_g = 0.768, s.e. = 0.076$) (Methods). The estimate of genetic correlation between body mass index (BMI) and CI from 24 h recall ($\hat{r}_g = 0.174, s.e. = 0.051$) was slightly lower after adjusting for sugar/sweetener added ($\hat{r}_g = 0.148, s.e. = 0.050$), suggesting a role of added sugar/sweetener in the associations between CI and health-related outcomes. This conclusion is further evidenced by the observation that people who drink coffee with added sugar/sweetener had a higher disease burden than those without (1.45 vs. 1.22, Supplementary Table 5). Besides the additives, beverage subtypes might also lead to differences. There are four subtypes of coffee reported from the UKB participants: decaffeinated, instant, ground, and others (data-field ID: 1508). We adjusted CI for the coffee subtypes and re-ran the MR analysis. The results were largely consistent but with an exception for VVLE (Supplementary Fig. 21). That is, before adjusting for coffee subtypes, the causal estimate was not significant for 4/8 methods, but after the adjustment all eight methods provided significant protective $b_{xy}$ estimates. These results suggest that part of the protective effect of CI on VVLE could be masked by the mixture of coffee subtypes. The estimated effects of CI on diseases remained almost unchanged after adjusting for urinary biomarkers including blood urea nitrogen levels, urinary albumin-to-creatinine ratio, and estimated glomerular filtration rate creatinine (Supplementary Fig. 22), indicating that the identified causal associations of CI are unlikely to be confounded by urinary or renal functions. To investigate the potential bias in the causal estimate that may be introduced by unmeasured confounders affecting the exposure-outcome relationship, we adopted the Latent Heritable Confounder MR (LHC-MR)[80] method. This allowed us to estimate both the bidirectional causal effects and the effects of potential confounders. In general, the estimates from LHC-MR aligned with those from other MR methods (Supplementary Data 13), except for allergic rhinitis and osteoarthritis (Supplementary Fig. 23 and Supplementary Data 8). We have also attempted to estimate the effects of CI/TI on common diseases excluding the top two IVs, and the results were mostly consistent except for those from MR-Egger (Supplementary Fig. 24).

This study has several limitations. First, our stratified regression showed that the health effects of TI and CI could be dosage-dependent, and the pattern also varied for different diseases. Thus, a triangulation framework that combines multiple methods is necessary to dissect the genetic and causal relationship between SUB and common diseases. Different MR methods have different underlying assumptions that may not be satisfied in every pair-wise association we tested. In this case, comparing multiple MR methods would be recommended to identify robust causal associations between modifiable risk factors and common diseases. Nevertheless, among the significant associations we identified, there was no scenario in which the $b_{xy}$ estimates from different methods were significant but in opposite directions, indicating the robustness of our findings. Second, despite identifying that the minor allele effects of the top two GWAS signals for TI/HTI oppose their effects on MTI (Table 1), we still cannot fully elucidate the discrepancy in the context of underlying biological mechanisms. There are more than 100 metabolites significantly associated with coffee intake[81]. The two top signals were linked to genes *CYP1A1* and *AHR*, both of which are associated with the caffeine degradation process[82,83]. Future studies are warranted to understand whether caffeine and/or other metabolites has a dose-dependent mechanism or whether the pattern we observed was just induced by potential confounders, such as substances added to coffee or tea.

In conclusion, this study combines different analytical frameworks to detect putative causal links between SUB and common diseases. Smoking showed widespread risk effects on common diseases and alcohol consumption showed risk effects specifically on cardiovascular and metabolic diseases. It was also highlighted that coffee and tea intake could exert dosage-dependent effects on several diseases and the underlying causes are complicated, possibly due to heterogeneous genetic architecture and confounding effects. The complexity of causal effects between SUB and common diseases should be interpreted with cautions, especially when significant differences exist in the causal estimates among different MR methods. Future studies with large-scale clinical diagnosed phenotypes such as nicotine dependence, alcohol use disorder, and caffeine dependence would be helpful to elucidate the genetic heterogeneity between habitual consumption and substance use disorder.

## Data availability

GWAS summary statistics of the seven SUB traits are available at https://yanglab.westlake.edu.cn/pub_data.html or https://doi.org/10.5281/zenodo.10596339[84]. All the data used in this study can be accessed by applying to the UKB. The individual-level original and pre-processed data cannot be directly shared due to restrictions set by the UKB. The numerical data underlying Figs. 1–4 can be found in Supplementary Data 2, 6, 8, and 9, respectively. The numerical data underlying Fig. 5 can be found in Supplementary Data 1–4, 6, 8, 9, and 11. All other data can be obtained from the corresponding author (or other sources, as applicable) upon reasonable request.

## Code availability

The GSMR/GSMR2 tools are integrated into the GCTA software package (v1.93.3), and the source code for GCTA v1.93.3 is available at https://yanglab.westlake.edu.cn/software/gcta/#GSMR (https://doi.org/10.5281/zenodo.5226943)[85]. The GitHub repositories for the GSMR2 and GSMR R packages can be found at https://github.com/jianyanglab/gsmr2 (https://doi.org/10.5281/zenodo.10595875)[86] and https://github.com/jianyanglab/gsmr (https://doi.org/10.5281/zenodo.10595809)[87], respectively. The code for the main analyses presented in this manuscript can be accessed at https://github.com/anglixue/MR_SUB (https://doi.org/10.5281/zenodo.10586538)[88].

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

## Acknowledgements

This research was supported by the "Pioneer" and "Leading Goose" R&D Program of Zhejiang (022SDXHDX0001 and 2024SSYS0032), the Leading Innovative and Entrepreneur Team Introduction Program of Zhejiang (2021R01013), the Australian Research Council (FT180100186 and FL180100072), the Australian National Health and Medical Research Council (1113400 and 1177268), and the Westlake University Research Center for industries of the Future (WU2022C002 and WU2023C010). This study makes use of data from the UK Biobank (project ID: 12505 and 66982). A full list of acknowledgements to the UK Biobank data can be found in the Supplementary Note 4. We thank for Jonathan Sulc and Zoltán Kutalik for their assistance and insightful discussions regarding the PolyMR analysis.

## Author contributions

J.Y. and A.X. conceived the study. J.Y., A.X., and J.Z. designed the experiment. A.X. performed all the analyses and simulations. A.X., J.Y., and Z.Z. contributed to the GSMR and HEIDI methodology development and software implementation. H.W. assisted in the simulation design and coding. L.J. curated the GWAS summary statistics from published studies. P.M.V. provided critical advice in data analysis and interpretation of the results. A.X., J.Y., and J.Z. wrote the manuscript with the participation of all authors. All the authors approved the final version of the manuscript.

## Competing interests

The authors declare no competing interests.
