## [Peer Review File · Communications Medicine]

This manuscript has been previously reviewed at another Nature Portfolio journal. This document only contains reviewer comments and rebuttal letters for versions considered at Communications Medicine

REVIEWERS' COMMENTS:

Reviewer #2 (Remarks to the Author):

1. Regarding the sample overlapping issue for the exposure and outcome GWAS in 2 sample MR analysis, I very much appreciate the authors performed the simulation analysis to investigate the issue. As shown in the new Supplementary Figure 23, the median b_{xy} is not biased with overlapping samples, which is expected. The sample overlapping is expected to affect the $SE(b_{xy})$, where $SE(b_{xy})$ is expected to be smaller when sample overlapping presents. It is quite obvious what's happening with sample overlapping as shown in Supplementary Figure 23, where the distribution of b_{xy} spread much narrower for the "full overlap" than "no overlap". This, in turn, also affects the test statistics where the median test statistic is high for "full overlap" than "no overlap" for most MR methods (8 out of the 11 methods by my eyeballing) in Supplementary Figure 23. The differences may not be significant in the simulation, but over large amount of real data MR analyses, this kind of difference may still produce false positive results in some cases.

This is actually a minor methodological point that should not affect the overall conclusion of the real data analysis part of this paper. However, this underestimate of $SE(b_{xy})$ due to overlapping samples in 2 sample MR is pretty established from methodology point of view (for any ratio estimate using delta method to derive $SE(b_{xy})$ including GSMR I believe). It just seems weird to me trying to contour a theoretical conclusion with simulations, for which results are entirely depending on the simulation set-up. I will leave it to the authors to decide whether they want to tone down their claim about the in-significance of this bias, as I could not give any contour argument to their simulation results. I just hope that this will not add to the more common practice of overlooking the sample overlapping issue in the MR literature.

2. Otherwise I have no further comments. The authors did a great job with a lot of additional analyses to address my comments.

Reviewer #3 (Remarks to the Author):

The authors have thoroughly addressed the multiple comments that I had made after the first round of reviews and I believe they did a tremendous job and the paper is much improved. I am strongly in favor of publishing the manuscript in Communication Medicine.

Best,

Marie Verbanck

We thank the two reviewers for their comments on the revised manuscript.

As per the editor's request, we have added a sentence about the motivation for comparing MR methods at the beginning of the first result section: "Prior research has indicated that some GWAS on substance use behaviours (SUBs) may be biased by potential confounders, leading to invalid IVs. This necessitates the re-evaluation of MR methods through simulation" (lines 230-232).

Below are our responses to the additional comments:

REVIEWERS' COMMENTS:

Reviewer #2 (Remarks to the Author):

1. Regarding the sample overlapping issue for the exposure and outcome GWAS in 2 sample MR analysis, I very much appreciate the authors performed the simulation analysis to investigate the issue. As shown in the new Supplementary Figure 23, the median b_{xy} is not biased with overlapping samples, which is expected. The sample overlapping is expected to affect the $SE(b_{xy})$, where $SE(b_{xy})$ is expected to be smaller when sample overlapping presents. It is quite obvious what's happening with sample overlapping as shown in Supplementary Figure 23, where the distribution of b_{xy} spread much narrower for the "full overlap" than "no overlap". This, in turn, also affects the test statistics where the median test statistic is high for "full overlap" than "no overlap" for most MR methods (8 out of the 11 methods by my eyeballing) in Supplementary Figure 23. The differences may not be significant in the simulation, but over large amount of real data MR analyses, this kind of difference may still produce false positive results in some cases.

This is actually a minor methodological point that should not affect the overall conclusion of the real data analysis part of this paper. However, this underestimate of $SE(b_{xy})$ due to overlapping samples in 2 sample MR is pretty established from methodology point of view (for any ratio estimate using delta method to derive $SE(b_{xy})$ including GSMR I believe). It just seems weird to me trying to contour a theoretical conclusion with simulations, for which results are entirely depending on the simulation set-up. I will leave it to the authors to decide whether they want to tone down their claim about the in-significance of this bias, as I could not give any contour argument to their simulation results. I just hope that this will not add to the more common practice of overlooking the sample overlapping issue in the MR literature.

Re: We thank the reviewer for raising this additional concern. To prevent any misinterpretation of simulation results, we have added a statement: "It is noteworthy that inflation in test-statistics due to sample overlap is a well-recognized issue in two-sample MR methods, and while the simulations in this study suggest that the primary conclusions are highly unlikely to be influenced by sample overlap, they should not be misconstrued as dismissing the issue of sample overlap entirely" (lines 165-168).

2. Otherwise I have no further comments. The authors did a great job with a lot of additional analyses to address my comments.

Re: We extend our gratitude to the reviewer for their comments on both the original and revised versions of the manuscript, which have significantly contributed to the improvement of the paper.

Reviewer #3 (Remarks to the Author):

The authors have thoroughly addressed the multiple comments that I had made after the first round of reviews and I believe they did a tremendous job and the paper is much improved. I am strongly in favor of publishing the manuscript in Communication Medicine.

Best,

Marie Verbanck

Re: We thank the reviewer for taking time to review our revised manuscript again. Her comments on the previous version of the manuscript have significantly contributed to the improvement of our paper.